# Sustainable Materials with Enhanced Mechanical Properties Based on Industrial Polyhydroxyalkanoates Reinforced with Organomodified Sepiolite and Montmorillonite

**DOI:** 10.3390/polym11040696

**Published:** 2019-04-16

**Authors:** Lidia García-Quiles, Ángel Fernández Cuello, Pere Castell

**Affiliations:** 1Tecnopackaging, Polígono Industrial Empresarium C/Romero N° 12, 50720 Zaragoza, Spain; 2University of Zaragoza, Escuela de Ingeniería y Arquitectura, Av. Maria de Luna, 3, 50018 Zaragoza, Spain; afernan@unizar.es; 3Fundación Aitiip, Polígono Industrial Empresarium C/Romero N° 12, 50720 Zaragoza, Spain

**Keywords:** biopolymers, nanoclays, nanobiocomposites, extrusion-compounding, polyhydroxyalkanoates, thermal properties, mechanical properties, differential scanning calorimetry, nuclear magnetic resonance, X-ray diffraction

## Abstract

Microplastics have become one of the greatest environmental challenges worldwide. To turn this dramatic damage around, EU regulators now want to ensure that plastic itself is fully recyclable or biodegradable. The aim of the present work is to develop a biobased and biodegradable biocomposite based on commercial polyhydroxyalkanoates (PHAs) and nanoclays, with the objective of achieving a reduction of rancid odour while avoiding any loss in thermomechanical properties, thus tackling two key disadvantages of PHAs. This research aims at completely characterising the structural, thermal and mechanical behaviour of the formulations developed, understanding the compatibility mechanisms in order to be able to assess the best commercial combinations for industrial applications in the packaging and automotive sectors. We report the development of nine nanobiocomposite materials based on three types of commercial PHA matrices: a linear poly(3-hydroxybutyrate) (P3HB); two copolymers based on poly(3-hydroxybutyrate)-*co*-poly(4-hydroxybutyrate) (P3HB-*co*-P4HB); and nanoclays, which represent a different polar behaviour. Dispersion achieved is highly relevant compared with literature results. Our findings show impressive mechanical enhancements, in particular for P3HB reinforced with sepiolite modified via aminosilanes.

## 1. Introduction

At least 8 million tonnes of plastics leak into the ocean each year, which is equivalent to dumping the contents of one bin truck into the ocean per minute. In 2016, 27.1 million tonnes of plastic waste were collected through official schemes in Europe, from which 31.1% of plastic post-consumer waste was recycled, 41.6% was dedicated to energy recovery and 27.3% was landfilled [1]. Moreover, landfill rates are very uneven across Europe. In countries where landfill bans are in effect (Belgium, Luxembourg, Netherlands, Germany, Denmark, Switzerland, Austria, Norway and Sweden), less than 10% of plastic waste is landfilled. In other countries, such as Spain and Greece, a staggering amount of over 50% of all plastic waste still finds its way to landfill [2]. Furthermore, the option of exporting plastic waste to EU or non-EU counties has been foreseen and allowed by the existing EU legislation given that there is evidence that recovery of materials is taking place under conditions that are equivalent to the EU legislation [3]. This situation is starting to change. For decades, China was the world’s largest importer of waste, but this has been changing after Beijing banned 24 types of scraps from entering its borders starting January 2018. This decision has forced other countries, such as Europe, USA and Japan, to improve management of their own waste [4]. The problem is even accentuated when talking about microplastics. It is estimated that between 75,000 and 300,000 tonnes of microplastics are released into the environment each year in the EU [5]. These are reasons why EU regulators now want to ensure that the plastic itself is fully recyclable or biodegradable.

In December 2015, the European Commission adopted an EU Action Plan for a circular economy, and in January 2018, the European Commission published its Communication ‘A European Strategy for Plastics in a Circular Economy’ as an ambitious step towards making the European plastics system more resource-efficient and to drive the change from a linear to a circular system [5]. A circular economy aims to keep products, components and materials at their highest utility and value at all times, emphasising the benefits of recycling residual waste materials [6]. The New Plastics Economy has three main ambitions:Create an effective after-use plastics economy by improving the economics and uptake of recycling, reuse and controlled biodegradation for targeted applications;Drastically reduce leakage of plastics into natural systems (in particular, the ocean);Decouple plastics from fossil feedstocks by exploring and adopting renewably sourced feedstocks.

Therefore, a new sustainable solution for the plastic sector needs to tackle three pillars: eco-design, functionality and end-of-life. In this sense, various investigations are aimed at decreasing the amounts of plastic waste and manufacturing products with less environmental impact via recycling strategies or via the use of biodegradable materials. Aware of the environmental impacts of the production of synthetic polymers from nonrenewable resources, a promising solution could be the usage of mixtures of biopolymers that gradually replace those synthetic polymers. It is relevant to recall that biopolymers, also referred to in some cases as bioplastics, can be classified into two main groups. When referring to their origin, they can be biobased or fossil-based, and when alluding to end-of-life, they can be biodegradable or nonbiodegradable. Different families and combinations can be found. For example, polylactic acid (PLA) as well as polyhydroxyalkanoates (PHAs) are renewable resource-based biopolyesters, in contrast to polycaprolactone (PCL), polybutylene succinate (PBS) and aliphatic–aromatic polyesters, which are petroleum-based biodegradable polyesters [7,8,9].

Nowadays, bioplastics represent about one percent of the approximately 320 million tonnes of plastic produced annually [10]. By 2020, biodegradable plastics are expected to represent 18% of bioplastics production and biobased, nonbiodegradable plastics will rise to 82% [11].

The development of new biopolymer materials will require availability of the raw material (second- and third-generation feedstocks), surpassing of market barriers facing economic disadvantages, and additional technological improvements dealing, for example, with higher heat resistance, UV stabilization, controlled barrier properties to water vapour and gases, and better mechanical properties. Blending is a useful strategy to modify material properties for specific applications. In addition to incorporation of fibres or nanoclays, the mentioned technical properties of biopolymers may be improved by chemical and physical crosslinking, or even with the use of surface treatments, such as grafting and coating. The economical barrier is expected to be compensated once disposal costs are taken into consideration and/or production volumes have increased further [12].

Polyhydroxyalkanoates (PHAs) are gaining attention in the biopolymers market due to their high biodegradation rates as well as processing versatility, thus representing a potential sustainable replacement for fossil oil-based commodities. PHAs are most relevant when referring to new biopolymer applications [13]. Polyhydroxyalkanoates (PHA) are polyesters which are intracellularly deposited by bacteria for energy storage. When carbon sources are alternated over time during the bacterial fermentation process, microorganisms synthesize PHA block copolymers. PHA biopolymers are formed mainly from saturated and unsaturated hydroxyalkanoic acids. Each monomer unit harbours a side-chain R group, which is usually a saturated alkyl group. These features give rise to diverse PHA combinations [14]. PHA biopolyesters include isotactic poly(3-hydroxybutyrate) P3HB, with a high melting point, being very crystalline and brittle; and the poly(3-hydroxybutyrate-*co*-valerate) PHBV copolymer, with a lower crystallinity and lower melting point. More recently, new customised copolymers have been developed by randomly incorporating controlled amounts of flexible linear aliphatic spacers along the main chain; for example, 3-hydroxypropionate (3HP) or 4-hydroxybutyrate (4HB) [15]. The results are semicrystalline copolymer structures designed to have a tailored melting point between 80 °C and 150 °C and that are less susceptible to thermal degradation during processing. Their properties range from brittle thermoplastics to gummy elastomers and can be controlled by the choice of substrate, bacteria and fermentation conditions. These biopolymers have attracted much interest for many new products in the medical and pharmaceutical sectors due to their natural ability to control drug release [16,17] and intrinsic biocompatibility properties [18]. More recently, the material has found new niches in the replacement of conventional oil-based plastic products. Good examples can be found in the food and cosmetic packaging sectors [19], mulch films for agricultural purposes [20], as bio-fuels [21] and even as interior automobile parts. However, PHA biopolymers face a technical barrier which is unique and intrinsic to this biopolyester family and the way it is produced: rancid odour [22]. Lipid residues and endotoxins often remain attached to the biopolymer after extraction. These lipids are oxidised to odourless and flavourless intermediates that could break into molecules giving off-flavours. These conditions are associated with the production of free radicals by autoxidation, which has been recognized as a potential shortcoming of PHA for many applications [23,24]. Different solutions can be approached in order to avoid polymer autoxidation. Most of them deal with the extraction and purification stages of the polymer, but these stages are the most expensive ones, especially if high purity is required. However, the current situation is that nowadays, plastic product converters purchase commercial-grade PHA with unsatisfactory smell, which is a handicap for many potential applications. Our proposed solution approaches the compounding stage of customised blends for industrial applications. It tackles the use of nanoclays with high adsorbance properties which are able to capture volatile compounds responsible for the displeasing fragrance.

Commercial PHAs purchased for plastic parts production are usually blended with other copolyesters and contain small amounts of plasticisers and metal elements, such as Na^+^, Ca^2+^ and Mg^2+^. These additives have an effect on the crystallisation, thermal stability [25], mechanical properties [26] and biodegradability of the commercial materials. However, this behaviour is not always functional enough for many industrial applications, and the use of other additives or fillers such as fibres [27,28,29] or nanoparticles [30,31] is needed. Their use gives rise to thermoplastic nanobiocomposite structures. To enhance the compatibility between the nanoparticle and the polymer is key to better fit the requirements of a certain application, although improving the compatibility of a heterogeneous system is often accompanied by the deterioration of other properties [32,33]. Sepiolite nanoclay fibres form ribbons with inner channels called zeolitic tunnels, offering interesting characteristics such as microporosity and large specific surface area. Due to its natural morphology, sepiolite is considered a good reinforcing agent as well as presenting an outstanding sorption capacity [34]. Moreover, there is much literature demonstrating that its surface functionalisation also helps to improve the transference of properties to the polymeric matrix [35]. In this work, natural sepiolite and sepiolite functionalised via aminosilanes are compared, and with montmorillonite as well. Montmorillonite falls into the smectite group and is largely used for its swelling and adsorption properties [36].

The objective of this research is to develop biobased and biodegradable biocomposites based on commercial PHA and nanoclays, enhancing their thermomechanical properties. This research aims at completely characterising the structural, thermal and mechanical behaviour of the formulations developed and tackles the understanding of the compatibility mechanisms that take place in order to be able to assess the best commercial combinations for industrial applications in the packaging and automotive sectors.

In the present work, three grades of PHAs were reinforced with modified and unmodified sepiolite and montmorillonite kindly provided by TOLSA (Spain). The clays differ in geometries, with sepiolites being T1 and T2 needles and montmorillonite being T3 fibres formed by sheets. The degree of improvement in the PHAs’ properties is a combination of the morphology of the clays, their dispersion in the polymer matrix and their interfacial polymer–clay interactions. The three candidates selected present different behaviours, from polar (T1) and neutral (T2) to nonpolar (T3) features, which directly affect the affinity for the PHA polyesters and therefore affects the matrix’s final properties in a different manner.

## 2. Materials and Methods

### 2.1. Materials

Three grades of PHAs were used as polymer matrixes: Mirel PHA1005, Mirel PHA3002 (both food contact-grade P3HB-*co*-P4HB grades, purchased from Metabolix USA) and Biomer PHB P226 (isotactic and linear short chain length scl-PHA homopolymer P3HB grade purchased from Biomer, Germany).

Three different modified and unmodified organoclays were kindly provided by TOLSA (hereby referred to as T1, T2 and T3, which have been previously characterised by other studies [34,37,38,39]), being: T1: Modified sepiolite: organically modified through aminosilane groups on the surface;T2: Natural sepiolite without modifications on its surface (naturally containing silanol groups, commercially marketed as Pangel 9);T3: Natural sodium montmorillonite (Na-MMT) modified with a quaternary ammonium salt; it is an anionic organoclay (highly compatible with nonpolar polymers).

### 2.2. Nanobiocomposite Preparation

PHA/clay formulations were prepared by extrusion-compounding with a 26-mm twin-screw Coperion ZSK 26 compounder machine (Germany). Twelve different formulations were studied in total, accounting for the three control matrixes plus nine developed materials prepared on the compounder machine by loading them with the three nanoclays at 3 wt.% in all cases (see description in Table 1). The melted polymers and nanoclay powder were mixed at a screw speed of 125 rpm; temperature was increased from 150 °C in the feeding zone up to 165 °C at the nozzle for PHA1005 and PHA3002 (P3HB-*co*-P4HB formulations) and slightly decreased from 140 °C in the feeding zone up to 160 °C at the nozzle when blending PHB226 (P3HB). The compounding was extruded through a 2-mm diameter die for a constant output of 10 kg/h. The extrudate was quenched in a water bath at room temperature, dried and cut into pellets.

Specimens for mechanical and tensile testing were obtained by injection moulding with a JSW 85 EL II electric injection machine (JSW, Tokyo, Japan) following ISO 178 and ISO 527 standards. Temperature profile was increased from 160 °C at the hopper up to 200 °C at the nozzle. Dosage and filling pressure were varied for each formulation injected. A packing pressure of 25% was applied.

### 2.3. General Characterisation Methods

Mechanical tests were conducted under ambient conditions using a Zwick Roell Z 2.5 (Zwick, Germany). At least five specimens per material were tested, according to ISO 178 and ISO 527 methodology.

Nuclear magnetic resonance (NMR) spectra were measured on a Bruker instrument at 25 °C using 400 MHz for the three neat biopolymers (PHB226, PHA1005 and PHA3002) (Bruker, Karlsruhe, Germany). Samples were dissolved in CDCl_3_ and washed to clean them up from mineral fillers. ^1^H NMR spectra were obtained. 

Thermal characterisation was carried out by differential scanning calorimetry (DSC) using a Mettler Toledo 223E (Mettler, Columbus, OH, USA). Dynamic heating was performed from room temperature to 220 °C at a rate of 10 °C/min for 8 mg samples placed into standard 40 μL aluminium crucibles, under a 100 mL/min flow of nitrogen. DSC tests were duplicated to ensure the reproducibility of results.

Energy-dispersive X-ray spectroscopy (EDX) was also used for samples’ chemical characterization in a Hitachi S3400N (Hitachi, Tokyo, Japan). The diffraction pattern was determined using a Bruker D8 X-ray diffraction (XRD) equipment using Cu Kα irradiation at 44 kV (Bruker, Karlsruhe, Germany). The diffractogram was carried out between 5° and 80° at a step of 0.5°/min. XRD was used to identify changes in PHA/clay blends’ crystalline structure. XRD tests were duplicated to ensure the reproducibility of results.

## 3. Results and Discussion

### 3.1. Mechanical Results (Flexural and Tensile)

The mechanical properties of composite materials are always a compromise between stiffness and toughness. These properties are generally mutually exclusive. The elastic modulus (E), tensile strength (σ_M_) and elongation at break (ε_B_) are useful parameters which describe the mechanical behaviour of the developed materials and are closely related to the internal microstructure. Toughness (U_T_) was calculated by integrating the stress–strain curves and obtaining the area under the curves. The mechanical properties determined from uniaxial tensile and flexural tests are summarized in Table 2.

It is well-documented that P3HB seems to be more crystalline, mechanically stiffer, stronger and less ductile than its copolymers [40]. According to Koller et al., pure P3HB presents a tensile strength of 40 MPa and 6% elongation at break, while pure P4HB presents 104 MPa and 1000%, respectively [41]. Cong et al. demonstrated that the addition of a 4HB copolymer at up to 30 wt.% into P3HB causes reductions in the storage modulus, stress at yield and stress at break, while the elongation at yield and at break increases [42]; however, our results show a different situation. It has to be taken into account that even our commercial-grade P3HB (PHB226) may contain small percentages of talc, plasticiser and other polyesters. The presence of these and other additives will be explored and discussed by DSC and NMR analysis. Therefore, the general statements and results obtained by other authors in similar research works where the polymers were synthetised and blended in a laboratory with nanoparticles might not always correspond to others’ findings. In addition, NMR results established the molar ratio of the P3HB-*co*-P4HB blends, with PHA3002 being the one containing the highest percentage of P4HB. The amounts of talc that each blend contains highly affects the elongation at break and toughness of the blends, and it is not possible to find a correlation due to its random behaviour.

Our results show that PHA1005 is the stiffest matrix with the highest elastic modulus under tensile and flexural stress, followed by PHA 3002 and finally by PHB226. However, PHB226 presents a high elongation at break, probably due to the PBA, and in consequence, it presents the highest toughness under tensile efforts. PHA3002 maintains an intermediate behaviour under tensile and flexural stresses.

A common behaviour found in many nanobiocomposites when nanoparticles are introduced is an increase in the elastic modulus, a preservation or even a slight increase in the tensile stress and a decrease in elongation at break [43]. Botana et al. demonstrated that the incorporation of small quantities of montmorillonite (2–10% in mass) with a certain degree of exfoliated structure have a great influence on the properties of the final material, such as mechanical strength, stiffness, thermal stability, conductivity and gas barrier properties [44]. Our samples comply partially with this generally observed behaviour, depending on the nanoclay reinforcing the matrix.

In the PHA1005 blends, the three nanoclays generally improved the stiffness of the material (both under flexural and tensile forces), with T3 (montmorillonite) being the one introducing the greatest enhancement. PHA1005_T3 shows a 35% higher Young’s modulus and 65% higher flexural modulus than neat PHA1005, although toughness was clearly compromised. Only for PHA1005_T2 was toughness enhanced under tensile forces, by 19%.

PHA3002 T1 (sepiolite modified via aminosilanes) greatly improves elongation at break, by 46%, and the increase in toughness by 69% compared to neat PHA3002 is therefore noticeable (although the elastic modulus is slightly compromised, falling by 13%). The modified surface of T1 may have acted as plasticiser with this matrix, favouring the interphase affinity. When testing the same material, PHA3002_T1, under flexural forces, an opposite behaviour was found, with the flexural modulus being increased and the elongation at break and toughness reduced. The authors consider that the alignment or orientation of the sepiolite ribbons with the flow when extruding the material may have also an important effect on the final strain and toughness. For PHA3002_T2, there was an increase of 31% in the Young’s modulus, while the elongation at break was maintained, as for PHA3002, and hence the tenacity was improved. The same tendency of behaviour was found for this material under flexural stress. Finally, a significant improvement was found when adding T3. The exfoliation of the layers (see XRD results) induced mechanical improvements for all parameters under tensile forces and was even more significant under flexural ones. PHA3002_T3 had its flexural modulus increased by 75%. The elongation at break was slightly reduced, but the improvement in stiffness was so high that final toughness was also increased by 10% compares to neat PHA3002.

For PHB226, the greatest improvement in mechanical properties is observed for T1, as it maintains the elastic modulus while substantially improving the elongation at break and therefore toughness. PHB226_T1 is the only material developed for which all the mechanical properties were maintained or increased. Under tensile stress, strain is increased by 48% and toughness by 55%, while under flexural stress, strain is increased by 9% and toughness by 17%. Probably the combination of a good dispersion in the blend, the large surface area of sepiolite and the organic modification of T1 produces a better interaction and affinity due to the aminosilane modification, which may present a better compatibility with the P3HB matrix.

This behaviour disappears for T2 (sepiolite without the functional modification). Despite the large amount of silanol groups, T2 acts as a filler, enhancing the stiffness of the material by 20% for tensile stress, but reducing elongation at break and toughness considerably. The material has similar mechanical properties to neat PHB226 under flexural stresses. Finally, when T3 is added, the overall stiffness of the material is improved, in particular increasing by 42% under flexural forces, but strain and toughness are compromised. The exfoliation of montmorillonite probably leads to the enhancement in toughness under tensile forces, but this is not as significant as the one produced by T1.

As a global pattern in all nanobiocomposites developed, T1 (modified sepiolite) probably has a better interphase, as elongation at break enhances while flexural modulus is maintained, whereas T2 formulations show higher rigidity and poorer toughness. T3 produces significant overall mechanical improvements, which may be induced due to exfoliation of clay layers leading to an increase of the effective aspect ratio, but probably the nonpolar behaviour of montmorillonite hinders a better polymer–matrix interaction.

Mechanical analysis demonstrates that PHA1005 is the matrix with the least overall improvement in mechanical properties when reinforcing the matrix with organoclays, while PHB226_T1 shows the greatest enhancement (modified sepiolite). However, it has to be taken into account that for all reinforced nanocomposites, there is an optimum load of filler, over which the matrix appears to be oversaturated. The combination of talc plus nanoclay may lead to this point being reached, which produces a loss of mechanical properties, especially regarding elongation at break and toughness. The results obtained for PHA1005 suggest that the material may contain larger amounts of talc than PHA3002.

Czerniecka-Kubicka et al. developed Biomer P3HB samples loaded with modified montmorillonite (cloisite 30B: natural montmorillonite modified with methylbis(2-hydroxyethyl)tallowalkylammonium cations) at 1 wt.%, 2 wt.% and 3 wt.% and evaluated the mechanical properties under flexural stress. The flexural modulus values found for nanocomposite containing 1 wt.% nanoclay increased by approximately 20% in relation to the nonmodified sample. Further increase of nanofiller content caused a decrease in flexural modulus values, but they were still higher than that of neat P3HB [45]. Our findings have shown an increase of 42% in flexural modulus when adding 3 wt.% of montmorillonite to Biomer P3HB (PHB226). The increase in the flexural modulus with PHA1005 and PHA3002 is even higher (65% and 75%, respectively). Therefore, we can confirm that either the dispersion or the surface modification has an extremely important effect on the composite. Dispersion is directly related to the extrusion-compounding process, where the temperature and shear force induced are key parameters in obtaining a homogenous blend without degrading the biopolymer. Compared to Czerniecka-Kubicka et al., our samples were mixed at lower temperatures, but at much higher rotation speeds (inducing higher shear and therefore favouring dispersion and delamination of montmorillonite in this case).

### 3.2. NMR

The ^1^H NMR spectra of P3HB (PHB226) and P(3HB-*co*-4HB) (PHA1005 and PHA3002) polymers is shown in Figure 1, with the various peaks labelled for the different protons in the 3HB and 4HB units, as well as for additional polyesters identified (PBA, plasticiser).

Very few articles have been dedicated to understanding the commercial grades of PHAs, which are the real ones that industry is incorporating in our daily products. Some characterisations have been done by Corre et al. [46] for commercial P3HB (PHB226), P3HB-*co*-P4HB (1006, 3002) and P3HB-*co*-3HV (Y1000P), but without explaining their composition in detail or giving further assessment on how to improve their weak properties.

Modification of PHA with plasticisers and other copolymers is a conventional technique for the improvement of the processability and brittleness of PHA. Nuclear magnetic resonance (NMR) is a useful analytical method to obtain information about the organic chemical structure of our blends.

The ^1^H NMR spectra of PHB226, PHA1005 and PHA 3002 were obtained. Results revealed the domain structure of P3HB in all samples. Specific peaks associated to P3HB and P4HB [47,48] protons were identified and the molar relation between PH3B and PH4B in PHA1005 and PHA3002 samples was obtained by integrating the peaks. The content found for P4HB corresponds to 23.5 mol % in PHA3002 and 17 mol % in PHA1005.

The presence of peaks different from P3HB were found in PHB226 samples (according to the product datasheet, it is 89.8% biobased P3HB). These are probably related to the addition of another biopolyester used as a plasticiser. Those peaks did not appear for PHA1005 or PHA 3002 polymers. Taking into account the nature of PHB, two candidates have been found to be potential copolymers in the blend: polybutylene adipate (PBA) components and triacetin (or citrate ester) [49,50]. Specific peaks that may be associated to PBA appeared between 1.3 and 1.7 ppm [48]. The presence of PBA is corroborated in DSC results, with the melting peak found at 50°C, which is characteristic of this polyester [51]. Initially, the addition of Polybutyrene adipate terephtalate (PBAT) was considered as it is a frequent additive used in many biopolymer formulations. However, the lack of a single peak at 8 ppm [43] confirms that no terephthalate had been added into any of the formulations. Often, these polyesters are added on purpose as plasticisers, and other times, they may be considered to be impurities from the purification stage of the PHA. It has to be taken into consideration that different batches of these materials may show slight differences due to additive traces used during polymer growth as carbon sources for the strain of microbe. For example, acetic acid, adipic acid, propionic acid or dodecanioic acid are used as precursors in the production of P3HB-*co*-P4HB or PHBHV4HB polymers [52]. The ^1^H-NMR spectrum of PHB226, PHA1005 and PHA3002 can be found in Figure 1.

### 3.3. DSC—Differential Scanning Calorimetry

DSC measurements were performed in order to observe the melting behaviour of the crystals and to determine the changes induced in the highly ordered structure. The DSC analysis of the samples was carried out over three cycles, involving a first heating cycle followed by a second cooling cycle and finally a third heating. This method ensures the removal of residual thermal behaviour in the polymers.

According to the literature data, the *T*_g_ of PHA biopolymers may vary between −1 and −48 °C, depending on the type and molar fraction of the second monomer (4HB, 3HV, 3HO, 3HHx) [53,54]. The *T*_g_ of P3HB has been reported to vary between −3 and 5 °C, and the *T*_g_ of P4HB at −46 °C [48]. Our thermograms for the neat matrixes are aligned to those of other neat PHAs reported by Corre et al. [46].

Comparing the *T*_g_ of our three raw matrixes, we can observe that the *T*_g_ of PHA1005 and the *T*_g_ of PHA3002 have almost no deviations around −24°C, which is about 10 °C lower than the *T*_g_ of PHB226. This behaviour is expected as the P4HB side groups in PHA1005 and PHA3002 increase the free volume in the molecule, resulting in a decrease of *T*_g_.

The increase in the *T*_g_ transition appreciated in the DSC diagram for PHA3002_3T1 should be highlighted, which might be induced by an increase of the amorphous phase in the interphase between the matrix and the nanoclay. For the case of PHA1005 bionanocomposite formulations, it can be observed that the three nanoclays (T1, T2 and T3) induce a decrease of *T*_g_, being especially remarkable in the case of T3. In addition, the transition becomes almost indiscernible, which suggests that nanoclays induce a plasticiser effect in the three cases. Moreover, *T*_g_ lowering could result from some interfacial interactions between the clay and the matrix producing disorganized molecular arrangements within the interphase, probably due to an agglomeration of filler, which might become a predominant factor for *T*_g_ decrease [55,56]. This tendency in behaviour can be observed for PHB226 too. T1 and T2 produce a clear decrease of *T*_g_. PHB226 is the polymer with the highest crystallinity in comparison with PHA1005 and PHA 3002. Thus, *T*_g_ results are difficult to be obtained as the transition is not clear in the DSC diagram due to the very low amorphous range. In the case of T3, it is not possible to find an approximate value. This is an indication about the good dispersion achieved with this nanoclay and is in coherence with the results obtained in XRD.

In Figure 2, we can observe that all diagrams show a clear crystallization peak, which indicates that our compounds undergo some small amount of crystallization while heating. Comparing the three matrixes, it can be corroborated that PHB226 is the one with highest crystallinity, followed by PHA3002 and finally PHA1005. This result is in contradiction to the findings of Bayari et al., who confirmed the fact that the degree of crystallinity of P(3HB-*co*-4HB) copolymers decreased with an increase in the amount of the 4HB content [57]. However, these are based on lab-produced PHA materials, not commercial blends, in which the use of additives tailors the crystallisation rates. Our results agree with the findings shown by Corre et al. [46], in which similar Mirel matrixes (P3HB-*co*-P4HB) under polarized optical microscopy (POM) featured larger spherulites with lower nucleation density than PHB226 (P3HB). The introduction of inorganic nanoparticles to increase the nucleation density and decrease the spherulite size is a common practice in commercial PHAs. Examples of these inorganic particles include tungsten disulphide inorganic nanotubes (INT-WS2), boron nitride (BN), talc (Mg_3_Si_4_O_10_(OH)_2_), hydroxyapatite (HA) and zinc stearate (ZnSt) [58], used as nucleation agents to modify the properties of P3HB-*co*-4HB. Wang et al. [58] suggested that the addition of talc increased the crystallisation degree of P3HB-*co*-4HB, but had little effect on the crystallisation rate. This cheap material is used at the industrial level by commercial material producers and it can be found in our neat PHB226, PHA1005 and PHA3002 matrices (according to EDX results carried out by the authors, in which Mg^2+^ cations and silicon are clearly identified).

The PHA1005 DSC diagram is accompanied by a rise in the cold crystallisation temperature and crystallisation enthalpy, with T1 being the nanoclay that induces the highest augmentation. Crystallization temperature (*T*_c_) increases by almost the same ratio in all cases (3–4%) for PHA3002 formulations when adding any of the three nanoclays with respect to the neat matrix. A similar tendency is obtained for PHB226 compounds, except for the case of PHB226_T3, which suffers a slight decrease with respect to neat PHB226. High Tc implies that the polymer crystallisation ability of the material is better [29].

The crystallinity of the samples was calculated for the second heating from the general equation: X_c_ (%) = (ΔH_m_/ΔH°_m_*(1 − w_t_))*100, where w_t_ is the clay fraction and AH°_m_ is the theoretical melting enthalpy of 100% crystalline PHB polymer, taken as 146 J/g [59].

P3HB chains typically form spherulites that are crystallised from the melt polymer. The nucleation density for P3HB is known to be excessively low, leading to the development of extremely large spherulites which grow radially within P3HB materials. The size varies from several micrometres to a few millimetres, depending on the crystallisation temperature and molecular weight [60]. Lamellar thicknesses in spherulites range from 5 nm to 10 nm, depending on the crystallisation temperature. It is well known that P3HB exhibits two crystal polymorphs: α and β crystals. It is assumed that the β-crystals appear from amorphous chains present between the lamellar crystals of the α-crystal (tie-chain). The β-form is introduced by the orientation of free chains in the amorphous regions between α-form lamellar crystals. The authors understand that these crystals form part of the so-called rigid amorphous phase (RAF) for this particular case of PHA structure. Di Lorenzo et al. assigned a peak around 45 °C to the RAF structure [61]. The presence or absence of this peak depends on the thermal history of the material. Our DSC results corresponds to the second heating, and therefore the thermal history of our matrixes has already been removed and the peak does not appear. However, our XRD results show the presence of β-crystals. The rigid amorphous structure growths during the first stage of cold crystallisation and slows down crystallisation before completion, creating an immobilised amorphous layer that surrounds the crystals. The physical state of the rigid amorphous fraction affects the crystallisation kinetics of P3HB. The crystal dimensions of P3HB and P4HB have been deeply characterised in the literature.

Our DSC results show the two peaks related to the two distinct populations of crystals (P3HB and P4HB) for all composite formulations. To understand the effects induced on crystallinity, it is important to take into account the differences in the kinetics and crystal formation in the copolymers between P3HB and P4HB, as well as the particular modifications that the nature of each nanoclay (with different polar affinity and structure) is introducing into the system. When adding the nanoclays, different behaviours can be observed according to the interphase created, which is directly related to the dispersion grade and interaction of the nanoclays with each polymer.

According to the literature [62], crystals related to P3HB exhibit a higher *T*_m_ than those corresponding to PH4B. Different authors report the Tm for pure 3HB to be 171–175 °C [57], while the Tm for pure 4HB appears at 56–58 °C [38]. The Tm for P3HB reported in the literature ranges between 162 °C and 197 °C [63]. The differences in Tm indicate that the size or thickness of P3HB crystals is greater than those for P4HB. Volova et al. compile quite a lot of information related to thermal behaviour and the structure of the different monomers and polymers that form PHAs, in particular for P3HB, P4HB, PHV and PHH [64].

In this research, PHA1005 and PHA3002 samples showed two endothermic peaks which correspond to the two crystalline phases: the 3HB-rich crystalline microregion at 167 °C and the 4HB-rich crystalline microregion around 157 °C. According to the literature [47], P(3HB-*co*-4HB) crystallises like P(3HB), with the 4HB units acting as defects in the crystal lattice when the 4HB content is less than 30 mol % (which has been confirmed in this study with the NMR results). Thus, the multiple melting behaviour of P(3HB-*co*-4HB) samples corresponding to PHA1005 and PHA3002 originates from microphase separation [65]. Kabe et al. studied the transition of spherulite morphology and measured the radial growth rate of spherulites in the blend of polyesters composed of P(3HB-*co*-3HH) and neat PHB with polarisation optical microscopy. They concluded that the radial growth rate of spherulites of neat P3HB was 0.25 mm/min, and complete crystallisation took about 5 min, while for the copolymer, they were 0.0008 mm/min and 9 h, respectively [66]. Therefore, it might be deduced that P3HB crystals appear to present the fastest radial growth rate.

In addition, PHB226 shows a melting peak at 50 °C, indicating the presence of another polymer or plasticiser, as suspected from the NMR results. Mohanty et al. described the presence of citrate plasticiser in this Biomer grade without providing further information [49]. Other authors consider that the material may contain small amounts of other copolymers, such as PBAT or PLA [48,67,68]. Our characterisation results (NMR and DSC) confirm that the formulation contains small amounts of PBA (with a Tm reported between 50 °C and 60 °C) [69], but not PBAT, as there is no trace of terephthalate. Anyhow, comparing the three neat matrixes, it can be observed that PHB226 presents the highest crystallinity of 79.5%, followed by PHA3002 with 44.2% and PHA1005 with 41.2%. Knowing that PHA3002 contains the highest P4HB/P3HB ratio, the crystallinity of PHA3002 should have been lower than that for PHA1005. However, both materials present high loads of mineral fillers (talc, according to EDX results) that most probably influence the crystallisation of the samples, varying the nucleation points and kinetics. The system becomes even more complex when the nanoclays are introduced. There is a decrease in the melting temperature (Tm) of the nanocomposites compared with pure matrixes, as the presence of the nanoclay seemed to induce crystal defects. This observation suggests the formation of smaller crystals with larger imperfections, which melt at lower temperatures [37].

In the case of the PHA1005 copolyester, when incorporating T1 (sepiolite modified with aminosilanes), an increase in crystallinity is achieved, while when introducing T2 (natural sepiolite), it seems not to affect the crystallisation of the original matrix. On the contrary, the tendency of T3 (Na-montmorillonite) is to decrease crystallinity slightly. For the matrix PHA3002, the introduction of sepiolites (T1 and T2) decreases the crystallinity of the material in a similar rate. Furthermore, when T3 is incorporated, the effect is even more acute, which suggests a global tendency for the rigidisation of the side chains when the nanoclays are dispersed. It can be observed that nanoclays particularly affect the peaks which correspond to a major concentration of P3HB crystals, so these are mostly hindering the rearrangement of short P3HB crystals. This result can be explained according to an induced modification in the crystallisation kinetics.

In PHB226, both Tm and Δ*H*_m_ decreases, and so does the global Xc. For both sepiolite nanoclays, a clear difference can be found. The superficial aminosilane groups of T1 seem not to be modified as much as natural sepiolite does, neither in the amount of crystals formed nor in their size. Therefore, the global crystallinity of the polymer is maintained. Some authors have demonstrated that nanoclay surfaces can be useful in providing nucleation points [38]. When PHB crystallises in the presence of the clay mineral particles, crystals could grow on the particle surfaces. In these cases, fillers (such as sepiolite, montmorillonite, cellulose nanowhiskers or fine lignin powder) reduce the energy barrier for polymer crystallisation and increase the nucleating density, originating smaller spherulites in higher number than in neat PHB [56]. We can find a slight decrease in Tm for all nanoclays, indicating a small reduction in size of the crystals formed. In addition, we can observe that T2 (natural sepiolite) may present a worse dispersion inside the blend, favouring the formation of agglomerates and therefore reducing the amount of crystals formed (as ΔH_m_ decreases with respect to neat PHB226 or PHB226_T1). The discontinuity of the silica sheets on the outer edges in sepiolite fibre leads to the presence of numerous silanol groups (Si–OH) at their external surface, which allows easy functionalisation based on their reaction with coupling agents such as organosilanes [39,70]. In T1 nanoclay, sepiolite has been grafted with aminosilane groups which are very stable, generating an organophilic clay that can be more easily dispersed in low-polarity polymers than unmodified clays [71]. Chemical covalent functionalisation is believed to counteract the stacking forces in the nanoparticles and to lead to debundling [72]. This effect may be caused by the intercalation of the attached moieties that finally results in a more effective dispersion within the polymeric matrix. Therefore, the affinity T1 inside the PHB226 may be scattered enough to keep a similar amount of nucleating points.

Anyhow, the most noticeable drop in *T*_m_ and ΔH_m_ is found for PHB226_T3. The exfoliation of montmorillonite may lead to a very high dispersion of the nanoclay. Exfoliation of T3 has been confirmed by XRD results. A good exfoliation should give rise into an increment in crystallinity, as it favours the creation of nucleation points. Nevertheless, the reaction can be so fast that crystals formed may be irregularly arranged, which increases the amorphous regions (RAF) between lamellae, and hence a decrease of the global crystallinity of the material [66], which means that it may hinder PHB226 chain movements by absorbing PHB226 segments on its surface [73]. Botana et al. studied the kinetics and dispersion of organically modified montmorillonites (among them, Na-montmorillonite) in PHB blends under polarised microscopy. The polarised optical micrographs indicated differences in the spherulite size [44]. In particular, Na-montmorillonite produced a large amount of spherulites, but being smaller in comparison to neat PHB. The appearance of disordered areas surrounding some spherulites corroborate that intercalated/exfoliated montmorillonite produces larger amorphous regions.

### 3.4. WAX—X-ray Diffraction

The XRD spectra of the samples revealed the crystallisation pattern of our PHA samples, which follows the trend of standard P3HB and P3HB-*co*-P4HB. P3HB exhibits two crystal polymorphs: α and β crystals. Diffraction patterns can be found in Figure 3.

The profile of PHB226 exhibits distinct diffraction peaks patterns of 2θ at 13.58, 17.03, 19.93, 21.5; 22.26, 25.69 and 30.69, corresponding to orthorhombic crystal planes (0,2,0), (1,1,0), (0,2,1), (1,0,1), (1,1,1), (1,2,1) and (0,1,2) [74], respectively. Sharp peaks at 2θ = 13.58 and 17.03 show typical α-crystals, while 2θ = 19.93 reveals that β-crystal structure also appears. This peak is also observed for PHA3002 with lesser intensity than PHB226, while for PHA1005, it disappears. This behaviour suggests that PHA3002 presents higher crystallinity than PHA1005, which is in coherence with the results obtained in DSC. For PHA1005 and PHA3002, a new peak not found for PHB226 is observed at 2θ = 27.13, which is attributed to the crystal plane (0,4,0) [75], indicating a preferential order of the copolymer blend P3HB-*co*-P4HB for this crystal structure.

In the neat polymers (PHB226, PHA1005 and PHB3002), X-ray diffraction patterns show three peaks at 2θ = 9.55, 19.07 and 28.71 that correspond to the addition of talc (usually used by manufacturers to control the nucleation of the polymer). To corroborate the chemical structure of the original clay included in the neat polymers, EDX was carried out. The results confirmed the incorporation of talc.

The diffraction patterns of T1 and T2 shows the expected structure for sepiolite organoclays, with the principle peak being at 2θ = 7.29, showing highest intensity. Results agree with those of Penning et al. as the basal interlayer distance of pristine sepiolite seems not to be altered by the silanisation process [76], and as the authors had corroborating calculations for the basal distance for d(1,2,1) and d(1,2,0), respectively (see Table 3). In addition, T3 presents three well-differentiated peaks at 2θ = 14.03, 19.83 and 61.9, which are typical of montmorillonite organoclay. These outcomes agree well with results previously reported in the literature [77,78,79,80,81].

The effects of each nanoclay on PHB226 and therefore on P3HB are different for each case. The peak associated with 2θ = 7.29 suffers from variations when blending the polymers with the nanoclays. T1 presents higher intensity than T2, which indicates that T2 presents a more effective interaction with the polymer. This result may be contradictory to the findings of DSC. At times, the greater the amount of organic modifier in clays, the greater the impediment to debundle (T2) or exfoliate (T3), and this seems to be the behaviour observed between T1 (modified) and T2 (natural). Moreover, for T3, the peak has completely disappeared, which may indicate the complete intercalation or exfoliation of the clay sheets, which was expected due to the laminar structure of this nanoclay. PHB226_T1 presents sharper peaks for 2θ = 13.58 and 17.03 than for PHB226_T2 and PHB226_T3 or even the neat PHB226, suggesting a preferred α-crystal orientation for planes (0,2,0) and (1,1,0) and a more ordered structure, which is aligned to the increase in crystallinity observed in DSC results (raising X_c_ by 5%). Moreover, a well-defined peak can be observed for the blend PHB226_T3 at 2θ = 19.93, indicating that T3 favours the introduction of the β form, which corroborates the creation of further amorphous regions due to the speeding up of crystallisation kinetics. In addition, the peaks corresponding to talc, 2θ = 19.07 and 28.71, had much lower intensity for PHB226T3, which suggests an intercalation with the talc of the PHB226 blend, producing a high dispersion of these mineral additives.

A global decrease of crystallisation for both P3HB-*co*-4HB blends (PHA1005 and PHA3002) can be observed, as not as many and such well-defined peaks can be found compared to PHB226 (higher-purity P3HB). Regarding PHA1005, 2θ = 13.58 and 17.03 are attenuated for PHA1005_T1 and are supressed for PHA1005_T2 and PHA1005_T3, indicating again the debundling and exfoliation of T2 and T3, respectively. For PHA3002, the addition of sepiolite T1 and T2 accentuates the α-form crystal formation, as again, the peaks at 2θ = 13.58 and 17.03 appear to sharpen compared to neat PHA3002 and PHA3002_T3. Often, if a peak is shifted to a lower angle or is reduced in intensity, it can be produced by the increase in the interplanar distance. It is usually an indication of good dispersion. The d-spacing calculated from the XRD figures for all samples is listed in Table 3. D-spacing for (P3HB-*co*-P4HB) apparently decreases when 4HB content is increased (PHA3002). These differences in basal distance can be appreciated in Table 3 for all diffracted angles and with particular attention at d(1,1,1) and d(1,2,1).

The basal distance has been increased for the main α-crystal planes in the case of T2 for PHB226 and PHA3002. Disappearance of the associated peak for PHA3002_T3 is also a relevant signal for a new order in the structure, probably induced due to exfoliation.

## 4. Conclusions

The present work demonstrates an industrial methodology to produce novel nanobiocomposite materials. Nine formulations were developed by adding 3 wt.% of sepiolite (T2), modified sepiolite via aminosilanes (T1) and montmorillonite (T3). Each nanoclay represents a different polar behaviour passing from one extreme to the other (polar, medium polarity and nonpolar). The different nanoclays were compounded into three types of commercial PHA matrixes: PHB226 (P3HB), PHA1005 and PHA3002 (P3HB-*co*-P4HB with different molar ratios of 4HB, being 17% and 23.5%, respectively). Results of NMR and EDX permitted us to better understand the composition of the commercial blends, as these contain talc, plasticisers, and even other polyesters. Therefore, we are dealing with complex quaternary and quinary composites.

For PHA1005 and PHA3002, two characteristic melting zones rich in 4HB and 3HB crystals are found at around 157 °C and 167 °C, respectively. A higher degree of crystallinity is observed for PHA1005 than for PHA3002, which a priori may seem contradictory. However, the results show the importance of nucleation kinetics which greatly affects the crystallisation process, as well as the appearance of RAF zones. The addition of nanoclays decreases Tm, which indicates the formation of smaller crystals. As a general conclusion, although minor exceptions appear, it can be said that T1 increases the Xc of the matrixes, T2 does not seem to modify it and T3 tends to decrease the overall Xc.

From the XRD patterns, we notice the appearance of α and β crystals typical of P3HB, as well as natural peaks proper tosepiolite, talc and montmorillonite. The appearance of β crystals confirms the formation of RAF zones, particularly for PHA1005. In addition, XRD confirms the exfoliation of montmorillonite, as well as the lack of complete debundling of both sepiolites. The grafting of aminosilane groups on top of the sepiolite surface is intended to favour the affinity and compatibility of the clay for the polymer; however, in this case, it acts as an impediment for dispersion. A good interaction between the nanoclay and the polymer is confirmed when mechanical properties are evaluated. The greatest mechanical improvement in terms of higher stiffness and toughness under tensile and flexural forces can be found for PHB226_T1. T3 produces significant overall improvement of the matrixes, but not as much as T1 does, and hence the polar/functional behaviour may predominate over dispersion to achieve good thermomechanical properties in complex polymer systems such as the ones selected. Anyhow, dispersion achieved in T1, T2 and T3 is highly relevant compared with literature results.

Our findings show impressive mechanical enhancements. Therefore, we believe that optimisation of the production parameters of the blend during extrusion compounding is critical to maximise the potential of any nanoparticle in the production of nanobiocomposites (being the lowest temperature as possible, medium to high screw speed and an low shear screw profile).

## Figures and Tables

**Figure 1 polymers-11-00696-f001:**
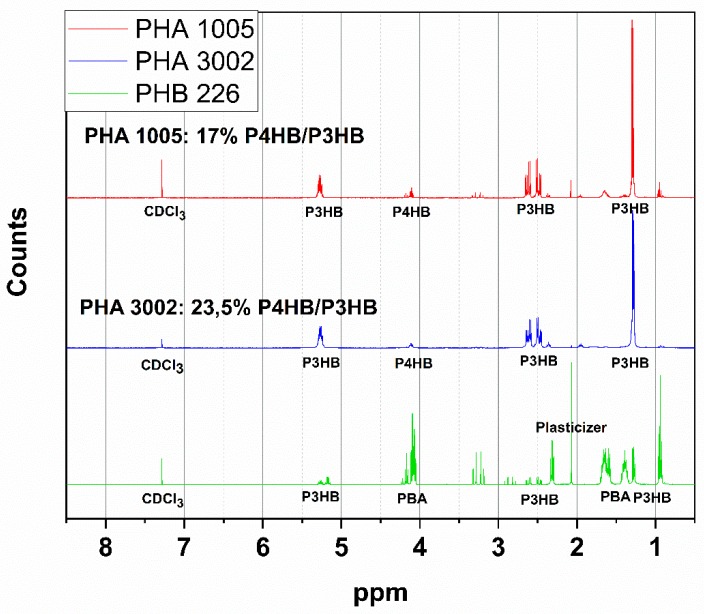
The ^1^H-NMR spectra of PHB226, PHA1005 and PHA3002.

**Figure 2 polymers-11-00696-f002:**
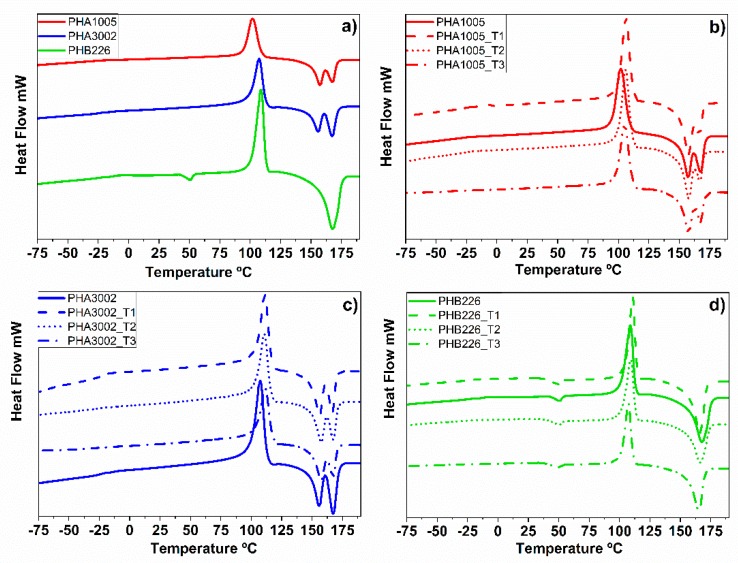
Differential scanning calorimetry (DSC) thermograms. (**a**) Raw matrices: PHA1005, PHA3002 and PHB226; (**b**) PHA 1005 loaded with 3 wt.% T1, T2 and T3; (**c**) PHA 3002 loaded with 3 wt.% T1, T2 and T3; (**d**) PHB 226 loaded with 3 wt.% T1, T2 and T3.

**Figure 3 polymers-11-00696-f003:**
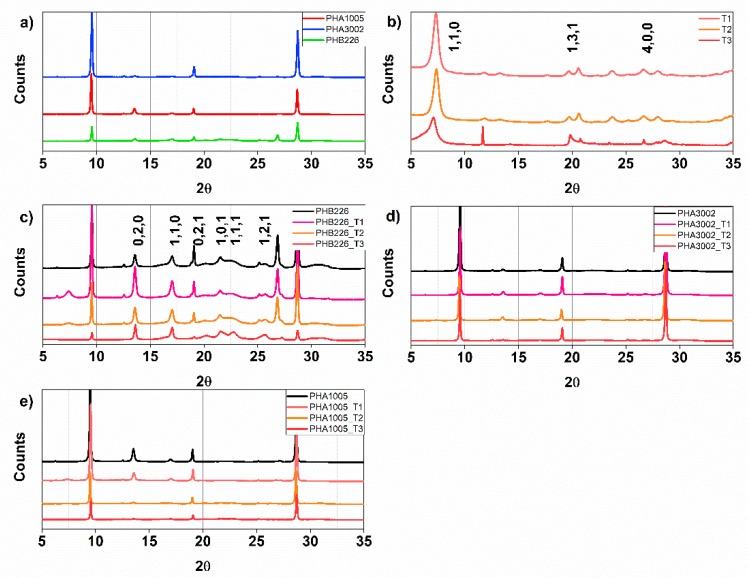
X-Ray diffraction patterns. (**a**) Raw matrixes: PHA1005, PHA3002 and PHB226; (**b**) Nanoclays: T1, T2 and T3; (**c**) Neat PHB226 and loaded with 3 wt.% T1, T2 and T3; (**d**) Neat PHA3002 and loaded with 3 wt.% T1, T2 and T3; (**e**) Neat PHA1005 and loaded with 3 wt.% T1, T2 and T3.

**Table 1 polymers-11-00696-t001:** Summary of material formulations based on poly(3-hydroxybutyrate) (P3HB); poly(3-hydroxybutyrate)-*co*-poly(4-hydroxybutyrate) (P3HB-*co*-P4HB) and nanoclays

Material Formulation	Commercial Matrix Used	Nature of the PHA	Type of Reinforcement (3 wt.%)
PHA1005	PHA1005	P3HB-*co*-P4HB	17% P4HB/P3HB and talc
PHA1005_T1	PHA1005	P3HB-*co*-P4HB	T1: Aminosilane sepiolite
PHA1005_T2	PHA1005	P3HB-*co*-P4HB	T2: Natural sepiolite
PHA1005_T3	PHA1005	P3HB-*co*-P4HB	T3: Sodium montmorillonite: quaternary ammonium salt
PHA3002	PHA3002	P3HB-*co*-P4HB	23.5% P4HB/P3HB and talc
PHA3002_T1	PHA3002	P3HB-*co*-P4HB	T1: Aminosilane sepiolite
PHA3002_T2	PHA3002	P3HB-*co*-P4HB	T2: Natural sepiolite
PHA3002_T3	PHA3002	P3HB-*co*-P4HB	T3: Sodium montmorillonite: quaternary ammonium salt
PHB226	PHB226	P3HB	Traces of PBA (polybutyladipate), plasticiser, and talc found
PHB226_T1	PHB226	P3HB	T1: Aminosilane sepiolite
PHB226_T2	PHB226	P3HB	T2: Natural sepiolite
PHB226_T3	PHB226	P3HB	T3: Sodium montmorillonite: quaternary ammonium salt

**Table 2 polymers-11-00696-t002:** Mechanical properties under tensile and flexural forces: modulus (E), flexural strength (σ_M_), elongation at break (ε_B_) and toughness (U) for all characterised materials.

Material	E (MPa)	σ_M_ (MPa)	ε_B_ (%)	U_T_ (MPa)	E_f_ (MPa)	σ_fM_ (MPa)	ε_fB_ (%)	U_fT_ (MPa)
PHB 226	2028 ± 68	21.8 ± 0.5	3.1 ± 0.3	53.9	1316 ± 28	30.2 ± 0.4	4.4 ± 0.2	98.8
PHB 226_T1	2004 ± 66	23.7 ± 0.6	4.6 ± 0.7	83.9	1306 ± 17	30.1 ± 1.0	4.8 ± 0.3	115.8
PHB 226_T2	2435 ± 88	22.1 ± 1.1	1.9 ± 0.3	33.6	1318 ± 68	34.1 ± 0.9	4.7 ± 0.4	111.5
PHB 226_T3	2006 ± 65	22.5 ± 0.3	4.2 ± 0.3	69.9	1871 ± 9	42.3 ± 0.3	3.6 ± 0.1	99.3
PHA 1005	2770 ± 99	22.4 ± 0.7	2.0 ± 0.1	32	1801 ± 39	31.5 ± 0.4	3.9 ± 0.1	92.8
PHA 1005_T1	3411 ± 98	25.2 ± 0.5	1.8 ± 0.1	33.9	2066 ± 71	32.2 ± 0.8	3.2 ± 0.1	71.9
PHA 1005_T2	3420 ± 218	25.9 ± 0.6	1.9 ± 0.2	38.2	2240 ± 88	33.9 ± 1.7	2.7 ± 0.1	60.3
PHA 1005_T3	3755 ± 41	23.8 ± 0.6	1.4 ± 0.1	24.9	2980 ± 90	47.6 ± 2.3	2.6 ± 0.2	81.6
PHA 3002	2263 ± 37	24.7 ± 0.5	2.8 ± 0.1	50	1342 ± 71	33.9 ± 1.5	5.0 ± 0.2	121.9
PHA 3002_T1	1997 ± 48	26.2 ± 0.3	4.1 ± 0.2	84.6	1485 ± 35	34.3 ± 1.1	4.2 ± 0.1	97.6
PHA 3002_T2	2961 ± 87	28.7 ± 0.3	2.9 ± 0.2	65.7	1807 ± 85	43.1 ± 1.1	4.4 ± 0.8	128.9
PHA 3002_T3	2463 ± 115	27.4 ± 0.2	3.3 ± 0.3	68.9	2358 ± 7	57.0 ± 0.7	4.3 ± 0.3	134.1

**Table 3 polymers-11-00696-t003:** Main crystallographic planes and d-spacing calculated from X-ray diffraction.

Material	D-spacing (Å)								
	(0,2,0)	(1,1,0)	(0,2,1)	(1,0,1)	(1,1,1)	(1,2,1)	(1,3,1)	(1,9,1)	(4,0,0)
PHB226	6.517	5.204	4.453	4.131	3.991	3.466	-	-	-
PHA1005	6.546	5.216	4.470 *	-	3.995	3.468	-	-	-
PHA3002	6.536	5.204	4.424 *	-	3.484	3.285	-	-	-
T1	-	12.120	-	-	-	-	4.311	2.565	3.344
T2	-	12.012	-	-	-	-	4.307	2.561	3.339
T3	-	12.444	-	-	-	-	4.282	2.565	3.342
PHB226_T1	6.514	5.203	4.439	4.060	3.953	3.463	-	-	3.317
PHB226_T2	6.524	5.203	4.389	4.137	3.976	3.465	-	-	3.320
PHB226_T3	6.505	5.197	4.368	4.118	3.912	3.470	-	-	3.268
PHA1005_T1	6.524	5.197	-	-	-	3.536	-	-	-
PHA1005_T2	6.542	5.227	-	4.118	3.929	3.545	-	-	-
PHA1005_T3	6.514	5.215	-	4.096	3.893	3.534	-	-	-
PHA3002_T1	6.514	5.197	-	-	-	3.534	-	-	3.322
PHA3002_T2	6.552	5.215	-	-	-	3.545	-	-	-
PHA3002_T3	6.524	* disappeared	-	4.104	-	3.534	-	-	-

* Almost not discernible/not found.

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
