# Peer review of "Sustainable Materials with Enhanced Mechanical Properties Based on Industrial Polyhydroxyalkanoates Reinforced with Organomodified Sepiolite and Montmorillonite"

_polymers, 2019, doi:10.3390/polym11040696_

Reviewer 1 Report

Dear Authors,

dear Editor,

This paper comprises some interesting results, fits the journal scope and certainly deserve publication. In my opinion the work is well done and interesting for the scientific community. Many composites were tested and the results are clear. The mechanical properties of the polymer are enhanced and the preparation of the composites is well described.

From the scientific point of view I have no objections.

Please enhance the quality of Figure 1, 2 and 3. Some words and letters are very hard to read.

Please add some perspectives opened by this study because of your interesting findings.

Author Response

We really thank reviewer 1 for his/her positive comments. Regarding the enhancement of figures we have proceed with the next modifications with the aim of improving the overall quality and ease their reading:

Figure 1 (NMR):

•             We have diminished the width of plot lines from 1 to 0,5 points, so that peaks can be clearly identified (single, double triple…).

•             We have increased the font size letter for peaks footprints.

•             We have reduced the horizontal axis scale from 0,5 to 8,5 instead of from 0 to 10 (just keeping 8ppm to demonstrate that not terephthalate is found).

•             We have increased the DPI resolution of the image from 600 to 1200.

Figure 2 (DSC):

•             We have improved some offsets to better identify the peeks.

•             We have increased the font size for both axes and used bold font.

•             We have decreased the font size of a), b), c) and d) markers in order to make the figure more homogenous.

•             We have increased the space between graphs, so that the horizontal axe information was easily to read.

•             We have increased the DPI resolution of the image from 600 to 1200.

Figure 3 (X-RAY):

•             We have decreased the font size of a), b), c), d) and e) markers in order to make the figure more homogenous.

•             We have increased the font size of crystallographic peaks.

•             We have increased the font size for both axes and used bold font.

•             We have increased the space between graphs, so that the horizontal axe information was easily to read.

•             We have increased the DPI resolution of the image from 600 to 1200.

Reviewer 2 Report

The paper describes the possibility to develop a biobased and biodegradable biocomposite based on commercial PHA and nanoclays. Impressive mechanical enhancements were reported.

The results are proper for publishing in this journal, however, there are issues that must be solved before its recommendation for publication, such as:

-          Please rephrase the last sentence from the Abstract in order to make it more understandable. I’m afraid that PHB226_T1sample does not tell anything to the reader.

-          Please, add recent studies performed by other groups at Introduction section, studies related with researches on polymer (PHA or other biopolymers)-clays (sepiolite, montmorillonite or others) for food packaging.

-          The introduction section is very well written and takes into account the existing plastic wastes problems. A small suggestions, the next paragraph doesn’t belong here especially that the present study is related (results obtained can be found in our publication: “Reducing off-flavour in commercial PHA grades  induced by autoxidation through the introduction of modified nanoclays”, which is under publishing evaluation). The paragraph brings you a little misleading. Eventually, explain what you have brought new in the presents study against the specified one (thermo-mechanical tests?). The study concerns the same compositions/samples, same types of clays?

-          At Material section please add more informations about T3. Did you used unmodified natural sodium montmorillonite (Na-MMT)? Because NaMMT clay is reacted with organomodifiers to make it an organoclay. Only in this manner it can be highly compatible with non-polar polymers, natural NaMMT being a hydrophilic clay. Please specify what type of organomodifier is about (long chain quaternary ammonium salts?, long chain alkoxysilanes?). Is it possible that the organomodifiers affect the Tg inducing the plasticizer effect that you mention about? Is it possible that the organomodifier to be degraded at the working extrusion temperature and to change the colour of the final materials? Please, consult some studies related with polymer-organomodified clays as Cl 30B, Cl 93A, Cl 15A and the effect of organomodifiers on the polymer matrix.

-          „In addition, T3 presents three well-differentiated peaks at 2θ = 14.03, 19.83 414 and 61.9 which are typical of montmorillonite organoclay. These outcomes agree well with results previously reported in the literature.” Please, specify the reference.

-          Section Materials: please delete the phrase ”But they were common reagents, so their specifications, manufacturers and others will not be detailed here”. I believe it is unsuitable.

-          Specify also if the results are reproducible and add some informations about duplicate/triplicate samples if there are.

-          Suggestion: if it is possible please add some TEM analyses in order to see the exfoliated/intercalated clay layers.

-          At Conclusion, please specify what you have brough new in comparison with other studies and please extract the main findings. I believe that the Conclusion section is too long and somehow repeats the Results section.

Therefore I would suggest publication of the paper after the minor revisions are taken into consideration.

With respect,

Author Response

The paper describes the possibility to develop a biobased and biodegradable biocomposite based on commercial PHA and nanoclays. Impressive mechanical enhancements were reported.

The results are proper for publishing in this journal, however, there are issues that must be solved before its recommendation for publication, such as:

We kindly appreciate the remarks and comments made by the Reviewer 2 for his/her encouraging comments. We answer to his/her specific comments/doubts below:

-          Please rephrase the last sentence from the Abstract in order to make it more understandable. I’m afraid that PHB226_T1sample does not tell anything to the reader.

Following this comment and a previous comment form the editor, this sentence has been modified in order to avoid using contracted references of the materials developed in the abstract, so that we do not confuse the reader as the material compositions have not been presented yet at abstract level. The sentence was reformulated as: “Our findings show impressive mechanical enhancements in particular for P3HB reinforced with modified Sepiolite via aminosilanes.”

-          Please, add recent studies performed by other groups at Introduction section, studies related with researches on polymer (PHA or other biopolymers)-clays (sepiolite, montmorillonite or others) for food packaging.

Thanks for the suggestion, we have search, read and added some new reference to increase the overall quality of the introduction references. These are:

8. Gao, S., Tang, G., Hua, D., Xiong, R., Han, J., Jiang, S., Zhang, Q., Huang, C. Stimuli-responsive bio-based polymeric systems and their applications. Journal of Materials Chemistry B 2019, 1-21. doi:10.1039/c8tb02491j

13. Ipsita; R., Visakh, P. M. Polyhydroxyalkanoate (PHA) Based Blends, Composites and Nanocomposites. Royal Society of Chemistry 2015, Doi:10.1039/9781782622314-FP001

32. Bumbudsanpharoke, N., Ko, S. Nanoclays in Food and Beverage Packaging – Review article, Journal of nanomaterials 2019, 1-13. Doi: 10.1155/2019/8927167

33. Ramos, Ó. L., Pereira, R. N., Cerqueira, M. A., Martins, J. R., Teixeira, J. A., Malcata, F. X., & Vicente, A. A. Bio-Based Nanocomposites for Food Packaging and Their Effect in Food Quality and Safety. Food Packaging and Preservation 2018, 271–306. doi:10.1016/b978-0-12-811516-9.00008-7

-          The introduction section is very well written and takes into account the existing plastic wastes problems. A small suggestions, the next paragraph doesn’t belong here especially that the present study is related (results obtained can be found in our publication: “Reducing off-flavour in commercial PHA grades induced by autoxidation through the introduction of modified nanoclays”, which is under publishing evaluation). The paragraph brings you a little misleading. Eventually, explain what you have brought new in the presents study against the specified one (thermo-mechanical tests?). The study concerns the same compositions/samples, same types of clays?

Authors thanks Reviewer 2 the comment and agrees that the reference to the autoxidation paper may be confusing. Therefore we have removed the reference to our other related paper. The new paragraph states as follows:

“The objective of this research is to develop biobased and biodegradable biocomposites based on commercial PHA and nanoclays enhancing their thermo-mechanical properties. This research aims at completely characterising the structural, thermal and mechanical behaviour of the formulations developed and tackles the understanding of the compatibility mechanisms that takes place in order to be able to assess the best commercial combinations for industrial applications in the packaging and automotive sectors.”

Regarding Reviewer 2 questions, yes, we have used the same list of composite materials for the autoxidation study. The global objective of the developed materials is twofold, reduce the odour of PHAs provoked by the release of oxidised volatiles thanks to its capture within the polymer due to the structure induced by nanoclays, and at the same time we are looking for a thermo-mechanical enhancement which is critical to ensure a good acceptance of these materials in the market.

-          At Material section please add more informations about T3. Did you used unmodified natural sodium montmorillonite (Na-MMT)? Because NaMMT clay is reacted with organomodifiers to make it an organoclay. Only in this manner it can be highly compatible with non-polar polymers, natural NaMMT being a hydrophilic clay. Please specify what type of organomodifier is about (long chain quaternary ammonium salts?, long chain alkoxysilanes?). Is it possible that the organomodifiers affect the Tg inducing the plasticizer effect that you mention about? Is it possible that the organomodifier to be degraded at the working extrusion temperature and to change the colour of the final materials? Please, consult some studies related with polymer-organomodified clays as Cl 30B, Cl 93A, Cl 15A and the effect of organomodifiers on the polymer matrix.

We thank the reviewer 2 for this comment and we agree that the modification of MMT might be not well explained. The Na-MMT used is a modified organoclay containing a quaternary ammonium salt. We have included it in the materials section, lines 166 and 167 of the manuscript.

Regarding reviewer 2 questions, we did not find changes in the polymer colour, so we don´t believe there was thermal degradation during extrusion-compounding. Authors have also checked literature related to the suggested commercial organoclays. The MMT for this article was specifically produced by TOLSA for this study and it is not exactly a commercial grade, but it has large similarities to Cl30B.

-          „In addition, T3 presents three well-differentiated peaks at 2θ = 14.03, 19.83 414 and 61.9 which are typical of montmorillonite organoclay. These outcomes agree well with results previously reported in the literature.” Please, specify the reference.

We agree with the reviewer that the reference is missed. We have included the references. These are:

80. Patrício, A. C. L., da Silva, M. M., de Sousa, A. K. F., Mota, M. F., & Freire Rodrigues, M. G. SEM, XRF, XRD, Nitrogen Adsorption, Fosters Swelling and Capacity Adsorption Characterization of Cloisite 30 B. Materials Science Forum 2012, 727-728, 1591–1595.doi:10.4028/www.scientific.net/msf.727-728.1591

 81. Xue, M.-L., Yu, Y.-L., Li, P. Preparation, Dispersion, and Crystallization of the Poly(trimethylene terephthalate)/Organically Modified Montmorillonite (PTT/MMT) Nanocomposites. Journal of Macromolecular Science, Part B 2010, 49(6), 1105–1116. Doi:10.1080/00222341003609385

 82. Sarier, N., Onder, E., Ersoy, S. The modification of Na-montmorillonite by salts of fatty acids: An easy intercalation process. Colloids and Surfaces A: Physicochemical and Engineering Aspects 2010, 371(1-3), 40–49. Doi:10.1016/j.colsurfa.2010.08.061

 83. Høgsaa, B., Fini, E. H., Christiansen, J. de C., Hung, A., Mousavi, M., Jensen, E. A., Pahlavan, F., Pederse, T.H., Sanporean, C.-G. A Novel Bioresidue to Compatibilize Sodium Montmorillonite and Linear Low Density Polyethylene. Industrial & Engineering Chemistry Research 2018, 57(4), 1213–1224. Doi:10.1021/acs.iecr.7b04178

 84. Krupskaya, V.V., Zakusin, S.V., Tyupina E.A., Dorzhieva, O.V., Zhukhlistov, A.P., Belousov, P.E., Timofeeva, M.N. Experimental Study of Montmorillonite Structure and Transformation of Its Properties under Treatment with Inorganic Acid Solutions, Minerals 2017, 7, 49. Doi:10.3390/min7040049

-          Section Materials: please delete the phrase ”But they were common reagents, so their specifications, manufacturers and others will not be detailed here”. I believe it is unsuitable.

The sentence has been removed during the last revision (from reviewers and editor). It is not found any more in the text.

-          Specify also if the results are reproducible and add some informations about duplicate/triplicate samples if there are.

Authors have included this suggestion at methodology section. Below the information added:

•     At least five specimens per material were tested according to ISO 178 and ISO 527 methodology. (186-187)

•     DSC tests were duplicated to ensure the reproducibility of results. (194)

•     XRD tests were duplicated to ensure the reproducibility of results. (199)

-          Suggestion: if it is possible please add some TEM analyses in order to see the exfoliated/intercalated clay layers.

Authors kindly appreciate this suggestion. However, we do not consider that including a TEM will significantly increase the detailed information given and explained in the paper. Therefore we have not included a TEM analyses for this paper. However authors take it into consideration for future publications.

-          At Conclusion, please specify what you have brought new in comparison with other studies and please extract the main findings. I believe that the Conclusion section is too long and somehow repeats the Results section.

From our warm point of view we believe that we may have not transmitted the novelty of our results properly. Our main findings can be summarized in three key points:

A better characterization and understanding of commercial PHAs, which are complex ternary blends.

Impressive mechanical results in the nanocomposites developed, and a better understanding of the interphase and the behavior of the nanoaclays within the matrix (being quaternary and      quinary composites).

·         Understanding the structure of the blends, how the nanolcays are dispersed and how the nature of the nanoclays affect the global crystallinity of the matrices, which in the end directly affects the thermal and remarkable mechanical results.

We agree with Reviewer 2 that the conclusions sections might be a bit long, however, we understand that it contains a good summary of the paper highlighting the key findings and that as it is written now, a reader can follow and understand the whole content of the paper easily.

Therefore I would suggest publication of the paper after the minor revisions are taken into consideration.

Authors truly appreciate it and again, thanks Review 2 to help us improving the quality of the article.

With respect,

Reviewer 3 Report

the paper is surely interesting: it is concernig the relevant topic of biodegradable plastic materials and presents a lot of experimental data.

the paper is not easy to read: I'd suggest to the authors to describe the results more schematically, in case adding  Tables summarizing the results.

Furthermore, in my opinion a scheme containing the chemical formulas of the different polymers and copolymers would be useful to achieve the interpretation of the spectra (in particular 1H NMR) and to try to understand the behavior and experimental data different from what reported in the literature for related systems.

A suggestion could be that a preliminar paragraph with the characterization of all the starting materials (clays and organic components, containing additional not expected components) would simplify the subsequent description of the results, in particular of those not expected on the basis of the literature.

the figures quality must be improved: in particular 1H NMR is difficult to study.

Even if the references are a lot, the book concerning PHA by RCS could be added ( Roy and Visakh eds. PHA based blends, composites and nanocomposites).

Author Response

the paper is surely interesting: it is concernig the relevant topic of biodegradable plastic materials and presents a lot of experimental data.

The authors truly appreciate all the comments and suggestions made by Reviewer 3 and thank the reviewer for his/her recommendation for publication. Please find detailed below the responses to his/her comments.

the paper is not easy to read: I'd suggest to the authors to describe the results more schematically, in case adding  Tables summarizing the results.

(See table in attached pdf file) We partially agree with Reviewer 3. We have been requested by the editor to re-organized part of the content of the article before sending it for your review. One of the suggestions was to delete a table summarizing the results coming from thermal analysis (DSC), which we understand is complex, as it was considered partially redundant with the related figure. We show this table removed below:

We will discuss this point with the editor, as suggested by the reviewer 3, and consider its introduction again in order to ease the follow up of the text.

Furthermore, in my opinion a scheme containing the chemical formulas of the different polymers and copolymers would be useful to achieve the interpretation of the spectra (in particular 1H NMR) and to try to understand the behavior and experimental data different from what reported in the literature for related systems.

A suggestion could be that a preliminar paragraph with the characterization of all the starting materials (clays and organic components, containing additional not expected components) would simplify the subsequent description of the results, in particular of those not expected on the basis of the literature.

We have included some further chemical description in the material´s section, in particular in Table 1 (organic components and additional not expected components), so that the reader can have pretty soon in the article an overall description about the molecules found during the characterization analysis. We expect this improvement helps to understand better the overall text.

The characterization of all materials is indeed the content of the article and we consider them part of the key main findings. The matrices are not in depth studied previously in literature. With regard to the nanoclays used, they are characterized in previous papers (one of them from previous work carried out by these authors) and hence it has been referenced in line 162. Only T3 is a non-commercial nanoclay and due to our confidentiality with TOLSA we are not allowed to publish deeper by the moment. However we have included the type of modification in Table 1 (a quaternary ammonium salt) and we consider it a competence to well know Cloisite 30.

the figures quality must be improved: in particular 1H NMR is difficult to study.

Regarding the enhancement of figures we have proceed with the next modifications with the aim of improving the overall quality and ease their reading:

Figure 1 (NMR):

·         We have diminished the width of plot lines from 1 to 0,5 points, so that peaks can be clearly identified (single, double triple…).

·         We have increased the font size letter for peaks footprints.

·         We have reduced the horizontal axis scale from 0,5 to 8,5 instead of from 0 to 10 (just keeping 8ppm to demonstrate that not terephthalate is found).

·         We have increased the DPI resolution of the image from 600 to 1200.

Figure 2 (DSC):

We have improved some offsets to better identify the peeks.

We have increased the font size for both axes and used bold font.

We have decreased the font size of a), b), c) and d) markers in order to make the figure more homogenous.

We have increased the space between graphs, so that the horizontal axe information was easily to read.

We have increased the DPI resolution of the image from 600 to 1200.

Figure 3 (X-RAY):

We have decreased the font size of a), b), c), d) and e) markers in order to make the figure more homogenous.

We have increased the font size of crystallographic peaks.

We have increased the font size for both axes and used bold font.

We have increased the space between graphs, so that the horizontal axe information was easily to read.

We have increased the DPI resolution of the image from 600 to 1200.

Even if the references are a lot, the book concerning PHA by RCS could be added ( Roy and Visakh eds. PHA based blends, composites and nanocomposites).

Thanks for the suggestion, we have search, read and added the suggested book as a new reference to increase the overall quality of the references and the paper.

13. Ipsita; R., Visakh, P. M. Polyhydroxyalkanoate (PHA) Based Blends, Composites and Nanocomposites. Royal Society of Chemistry 2015, Doi:10.1039/9781782622314-FP001

Reviewer 4 Report

The submitted work aimed to develop a biobased and biodegradable biocomposite based on PHA and nanoclays. The microstructure and thermal and mechanical properties of the biocomposites were comprehensively investigated. The data presented are of high quality and the manuscript can be accepted for publication after adding some sort of discussion to compare the current system with the state of the art. For example, in the introduction part, the author stated that “In this sense, various investigations are aimed at decreasing the amounts 66 of plastic waste and to manufacture products with less environmental impact via recycling strategies 67 or via the use of biodegradable materials……and aliphatic-aromatic polyesters, which are petroleum-based biodegradable polyesters.” I would like to suggest the author cite some newly published paper related to the biodegradable polymers (like PLA, PBS, PPC, etc. ) to make the reader understand the background more clearly, e.g., Chemical Engineering Journal, 2017, 307, 1017-1025; Carbon, 2016, 105, 305-313; Composites Part B: Engineering, 2017, 123, 112-123; Composites Science and Technology, 2018, 159, 171-179; Journal of Materials Chemistry B, 2019, 7, 709-729; Macromolecular Materials and Engineering, 2018, 1800336; Macromolecular Materials and Engineering, 2017, 302(1), 1600353; Polymer Chemistry 2018, 9(20), 2685-2720, etc. Besides, though the figure can be seen in the current version, the author should be better to provide more high-resolution figures in the revised manuscript in order to the meet the high demand of the journal of Polymers. For example, Figures 2 and 3 are not clearly and seems even a bit fuzzy. Lastly, though the manuscript is well written, I still suggest before it is considered to be accepted, all the sentences should be carefully checked to make sure it can be read with pleasure.

Author Response

The submitted work aimed to develop a biobased and biodegradable biocomposite based on PHA and nanoclays. The microstructure and thermal and mechanical properties of the biocomposites were comprehensively investigated. The data presented are of high quality and the manuscript can be accepted for publication after adding some sort of discussion to compare the current system with the state of the art.

The authors truly appreciate all the comments and suggestions made by Reviewer 4. Please find detailed below the responses to his/her suggestions. The manuscript has been modified taking them into account.

For example, in the introduction part, the author stated that “In this sense, various investigations are aimed at decreasing the amounts 66 of plastic waste and to manufacture products with less environmental impact via recycling strategies 67 or via the use of biodegradable materials……and aliphatic-aromatic polyesters, which are petroleum-based biodegradable polyesters.” I would like to suggest the author cite some newly published paper related to the biodegradable polymers (like PLA, PBS, PPC, etc. ) to make the reader understand the background more clearly, e.g., Chemical Engineering Journal, 2017, 307, 1017-1025; Carbon, 2016, 105, 305-313; Composites Part B: Engineering, 2017, 123, 112-123; Composites Science and Technology, 2018, 159, 171-179; Journal of Materials Chemistry B, 2019, 7, 709-729; Macromolecular Materials and Engineering, 2018, 1800336; Macromolecular Materials and Engineering, 2017, 302(1), 1600353; Polymer Chemistry 2018, 9(20), 2685-2720, etc.

Authors have looked for and read the articles and books suggested by Reviewer 4. We consider that on the one hand some of them can improve the introduction and frame of the articles. On the other hand, some other proposed articles are too much focused on nanofibers and carbon nanotubes which might be out of scope. Therefore we have done a selection of those considered more interesting by authors and included them as references. These are:

7. Kuang, T., Ju, J., Yang, Z., Geng, L., & Peng, X. A facile approach towards fabrication of lightweight biodegradable poly (butylene succinate)/carbon fiber composite foams with high electrical conductivity and strength, Composites Science and Technology 2018, 159, 171–179. doi:10.1016/j.compscitech.2018.02.021

8. Gao, S., Tang, G., Hua, D., Xiong, R., Han, J., Jiang, S., Zhang, Q., Huang, C. Stimuli-responsive bio-based polymeric systems and their applications. Journal of Materials Chemistry B 2019, 1-21. doi:10.1039/c8tb02491j

9. Kuang, T., Chang, L., Chen, F., Sheng, Y., Fu, D., & Peng, X. (2016). Facile preparation of lightweight high-strength biodegradable polymer/multi-walled carbon nanotubes nanocomposite foams for electromagnetic interference shielding. Carbon, 105, 305–313. doi:10.1016/j.carbon.2016.04.052

 Besides, though the figure can be seen in the current version, the author should be better to provide more high-resolution figures in the revised manuscript in order to the meet the high demand of the journal of Polymers. For example, Figures 2 and 3 are not clearly and seems even a bit fuzzy.

We deeply thank the Reviewer 4 for this appreciation. Regarding the enhancement of figures we have proceed with the next modifications with the aim of improving the overall quality and ease their reading:

Figure 1 (NMR):

·         We have diminished the width of plot lines from 1 to 0,5 points, so that peaks can be clearly identified (single, double triple…).

·         We have increased the font size letter for peaks footprints.

·         We have reduced the horizontal axis scale from 0,5 to 8,5 instead of from 0 to 10 (just keeping 8ppm to demonstrate that not terephthalate is found).

·         We have increased the DPI resolution of the image from 600 to 1200.

Figure 2 (DSC):

We have improved some offsets to better identify the peeks.

We have increased the font size for both axes and used bold font.

We have decreased the font size of a), b), c) and d) markers in order to make the figure more homogenous.

We have increased the space between graphs, so that the horizontal axe information was easily to read.

We have increased the DPI resolution of the image from 600 to 1200.

Figure 3 (X-RAY):

We have decreased the font size of a), b), c), d) and e) markers in order to make the figure more homogenous.

We have increased the font size of crystallographic peaks.

We have increased the font size for both axes and used bold font.

We have increased the space between graphs, so that the horizontal axe information was easily to read.

We have increased the DPI resolution of the image from 600 to 1200.

Lastly, though the manuscript is well written, I still suggest before it is considered to be accepted, all the sentences should be carefully checked to make sure it can be read with pleasure.

We thank the Reviewer 4 for this recommendation. Authors have read through the whole document during the review exercise refining some wording and completing some tables. We believe that with all changes introduced (thanks to reviewers suggestions), the quality of the article has increased considerably making easier its reading.